# The Role of Competition Area and Training Type on Physiological Responses and Perceived Exertion in Female Judo Athletes

**DOI:** 10.3390/ijerph19063457

**Published:** 2022-03-15

**Authors:** Ibrahim Ouergui, Slaheddine Delleli, Hamdi Chtourou, Damiano Formenti, Ezdine Bouhlel, Luca Paolo Ardigò, Emerson Franchini

**Affiliations:** 1High Institute of Sport and Physical Education of Kef, University of Jendouba, Kef 7100, Tunisia; ouergui.brahim@yahoo.fr (I.O.); sdelleli2018@gmail.com (S.D.); 2Institut Supérieur du Sport et de l’Education Physique de Sfax, Université de Sfax, Sfax 3000, Tunisia; h_chtourou@yahoo.fr; 3Activité Physique, Sport et Santé, UR18JS01, Observatoire National du Sport, Tunis 1003, Tunisia; 4Department of Biotechnology and Life Sciences, University of Insubria, 21100 Varese, Italy; damiano.formenti@uninsubria.it; 5Laboratoire de Physiologie de l’exercice et Physiopathologie, de L’intégré au Moléculaire “Biologie, Méde-cine, Santé”, UR12ES06, Faculty of Medicine Ibn El Jazzar, University of Sousse, Sousse 4000, Tunisia; ezdine_sport@yahoo.fr; 6Department of Neurosciences, Biomedicine and Movement Sciences, School of Exercise and Sport Science, University of Verona, 37131 Verona, Italy; 7Martial Arts and Combat Sports Research Group, School of Physical Education and Sport, University of São Paulo, São Paulo 05508-030, Brazil; efranchini@usp.br

**Keywords:** female, heart rate, blood lactate, combat sport, rating of perceived exertion

## Abstract

This study investigated the combined effects of competition area (4 × 4, 6 × 6, and 8 × 8 m) and judo-specific training type (tachi-waza, ne-waza, and free randori) on physiological responses and perceived exertion in female judo athletes. In a within-subject design, 12 female subelite and elite athletes who competed at regional or national levels with a mean training background of 8.4 ± 0.5 years performed the experimental conditions (i.e., combats (viz., matches) featuring different area/training type combinations) in random order. The following measurements at different time points were chosen: blood lactate before and after each match; heart rate before, mean, and peak for each match; and rating of perceived exertion immediately after each match. Two-factor analysis of variance was used to compare between conditions, while Bonferroni post hoc test and magnitude of difference were used to measure significance. There was no main effect of training type or area size on lactate before each match, heart rate (HR) before each match, HR mean during each match, and rating of perceived exertion. Main effects of training type and area size were found for lactate after each event, with the values being greater in free randori compared to tachi-waza and ne-waza and in 4 × 4 m compared to 6 × 6 and 8 × 8 m area. Main effects of training type and area size were also found in peak heart rate, with lower values in ne-waza compared to free randori and tachi-waza and in 8 × 8 m compared to 4 × 4 m area. The results demonstrate that varying training modality and area size may alter physiological responses during female judo combats by putting stress on the cardiovascular system and increasing anaerobic glycolysis solicitation.

## 1. Introduction

Judo is a high-intensity intermittent grappling combat sport requiring high physiological demands and technical–tactical excellence to be successful in competition [1]. Judo matches are typically played in an area ranging from 8 × 8 to 10 × 10 m [2]. Officially, a match lasts 4 min, with high-intensity periods interspersed by low-intensity movements, resulting in effort-to-rest ratios of 2:1 and 3:1, respectively [3,4]. Judo scores are calculated via the execution of throwing, immobilization, strangle, and joint lock techniques. Throwing techniques can result in ippon (full point) when the opponent is thrown on his/her back with strength, speed, and control or waza-ari (half point) when one of the aforementioned items is lacking. Ippon is also achieved when an immobilization is applied for 20 s or when the opponent gives up due to a strangle or joint lock technique. Waza-ari can also be achieved when an immobilization occurs for more than 10 s up to 19 s [5]. To maintain such high-intensity actions (i.e., scoring actions), the phosphagen and glycolytic energy systems are solicited, whereas low-intensity actions and the recovery process between high-intensity bouts are predominantly supported by the oxidative system [2,3].

To cope with the combat demands, judo-specific training modalities are highly recommended and considered as the best strategy to prepare athletes [6]. Throwing technique execution (i.e., nage-komi), repetitive technical training (uchi-komi), and standing and groundwork fight simulation (i.e., randori) are among the most common training modes used by advanced judo athletes [6,7]. These judo-specific exercises are useful to maximize training response and competitive performance as they can address both technical–tactical and physical components depending on the applied manipulation [7].

Moreover, judo matches can be disputed either in a standing position (i.e., tachi-waza) or on the ground (i.e., ne-waza [2]), which may require different physiological demands depending on the position. Therefore, understanding and characterizing the physiological responses induced by these different combat situations may be of interest from both a scientific and a coaching perspective. In this regard, Sikorski [8] compared a 5 min judo match disputed in tachi-waza with a match of the same duration performed in ne-waza. Higher blood lactate concentration [La] was found in the standing (11.3 ± 3.6 mmol·L^−1^) compared to the groundwork match (7.7 ± 2.2 mmol·L^−1^), which suggests lower glycolytic demands when combat is performed on the ground [8]. In a more recent study, Đerek et al. [9] investigated the effects of different training methods on muscle damage in young judo athletes (cadets). Specifically, the authors analyzed the effects of four practice bouts lasting 4 min each with 1 min intervals in between on alanine aminotransferase (ALT), aspartate aminotransferase (AST), creatine kinase (CK), and lactate dehydrogenase (LDH) concentrations. The authors showed the same increases in AST, CK, and LDH concentration after both tachi-waza and ne-waza sessions, suggesting similar muscle damage responses [9].

During national and international training camps, the number of athletes is often high, and the competitive area for each pair of athletes is restricted. Additionally, to enhance the physiological responses directly related to success in judo competition [1], restricting the competitive area seems to be an effective strategy to increase the intensity of the fight [10]. Reduced area size has been found to induce sufficient cardiovascular conditioning in different combat sports, such as kickboxing [10], and higher psychological stress in taekwondo [11]. For judo, combining time structure and area size has been found to modify the physiological responses and perceived exertion in female athletes [12]. Specifically, after-combat [La] values were greater post 4 × 4 m compared to 6 × 6 and 8 × 8 m area, and perception of effort results were greater post 4 × 4 m compared to 8 × 8 m area size. This provides evidence of the higher physiological demands in smaller competitive areas. Although training in smaller areas has been shown as a valid solution to elicit physiological responses, it is unresolved whether a combination of area size and training type would further impact the physiological demands in judo. To the best of the authors’ knowledge, no studies have investigated the physiological effects of manipulating area size combined with different specific training types in judo athletes. In the light of these considerations, this study aimed to verify whether the area size (i.e., 4 × 4, 6 × 6, and 8 × 8 m) and training type (tachi-waza, ne-waza, and free randori) can affect the physiological responses and perceived exertion in female judo athletes. It was hypothesized that varying the area and training type would affect the physiological responses. Specifically, based on the findings of a previous study [12], which focused only on physiological but not perceived exertion aspects, it was anticipated that smaller area and free randori would result in higher physiological strain. However, as judo athletes appear to be able to regulate their efforts during randori [13], it was also hypothesized that the rating of perceived exertion scores would remain unchanged.

## 2. Materials and Methods

To examine the combined effects of competitive area and judo training type on physiological responses and perceived exertion in female judo athletes, a within-subject design was applied considering 3 different areas and 3 training types, i.e., 9 different combinations. G*Power software (version 3.1.9.2, Kiel University, Kiel, Germany) was used to calculate the required sample size with α set at 0.05 and power (1-β) set at 0.80. According to Ouergui et al. [11], the target effect size was set at 0.40 (medium effect) primarily based on rating of perceived exertion results from that study. Thus, based on the aforementioned factors and data, the sample size required for the present study was 9. A total of 12 female judo athletes volunteered to participate in this study (age: 17.1 ± 0.4 years, height: 160 ± 7 cm, body mass: 59.8 ± 11.5 kg, body fat: 25.7 ± 5.5%, and judo experience: 8.4 ± 0.5 years). Note that the data were obtained as part of a previously published study by Ouergui et al. [12], who investigated the influence of different effort-to-rest ratios on physiological responses. It is important to emphasize that the athletes were the same. However, they executed judo matches with different area size and effort-to-rest ratio in the previous study, whereas they executed judo matches with different area size and training mode in the present study.

All athletes were grouped according to their weight categories and had been participating regularly in regional and national judo tournaments for more than two years. They were also assigned to the same training regimen 3 to 5 times per week (2 h per session). Athletes did not have any medical restrictions during the experimental period and refrained from any strenuous exercises in the 48 h before the experimental sessions. This study was conducted according to the latest version of the Declaration of Helsinki and was approved by the local research ethics committee (ethical approval number 2018-675). All participating athletes and their guardians and coaches were provided with a thorough explanation of the aims, benefits, and potentials risks of the study and signed a written informed consent thereafter.

Blood samples were taken from the fingertip 10 min before and immediately after each condition, and [La] was assessed using a Lactate Pro2 Analyzer (Arkray, Tokyo, Japan). [La] before and after the experimental conditions were used for further analyses. Before each condition, [La] was measured only to assess the same intergroup precondition value. The heart rate was measured before and every 5 s throughout the judo combat sessions (Polar Team2 Pro System, Polar Electro OY, Kempele, Finland), and HR before session (HRpre), mean (HRmean), and peak (HRpeak) values were used for the analyses. After being familiarized with the scale, athletes reported their rating of perceived exertion (RPE) in terms of Borg CR-10 scores (e.g., score 0 = nothing at all and score 10 = extremely strong [14]) after each combat session.

The procedures are described in detail in Ouergui et al. [12]. This research was performed during the preseason but out of the menstruation days. Seven days before the beginning of the research, the judo athletes were familiarized with the tests and the order of the randori sessions. Moreover, in a preliminary session, athletes accomplished the 20 m multistage shuttle-run test [15] to assess their maximum heart rate (HRmax). To minimize potential diurnal variation on the performance, experimental sessions were conducted at the same time of day and were completed during a maximum period of 30 days with at least 48 h of recovery (but not more than 72 h) between 2 subsequent sessions [12]. All experimental sessions were randomly administered (i.e., each athlete underwent all 9 different experimental conditions in a different order to minimize any possible practice effect and carryover effect on the outcomes) and directed by two investigators ensuring the safety of judo athletes and providing the same verbal encouragement. Each athlete had a different partner for each condition. Furthermore, all athletes were exposed to the same 4 min match duration and were instructed to continue the combat even when an ippon was scored [12]. Before each randori, athletes performed a standardized warm-up session lasting 15 min consisting of jogging and dynamic stretching, followed by 3 min of passive rest. Thereafter, baseline measures were performed.

The results are shown as means and standard deviations. Statistics was analyzed with SPSS 20.0 (IBM, Armonk, NY, USA). Normality of results was assessed and confirmed by means of the Kolmogorov–Smirnov test. Sphericity was assessed and confirmed by means of the Mauchly test. Data were analyzed using a 3 (competition area, i.e., 4 × 4, 6 × 6, and 8 × 8 m) × 3 (judo-specific exercise type, i.e., tachi-waza, ne-waza, and free randori) two-factor analysis of variance (ANOVA) with repeated measurements on the two factors. In case of significant main effects of competition area or judo-specific exercise type or interactions thereof, the Bonferroni corrected post hoc tests were computed as pairwise comparisons. The magnitude of differences between variables was interpreted using standardized effect size (Cohen’s *d*) and classified according to Hopkins [16]: *d* ≤ 0.20 (trivial); 0.20 < *d* ≤ 0.60 (small); 0.60 < *d* ≤ 1.20 (moderate)*;* 1.20 < *d* ≤ 2.0 (large); 2.0 < *d* ≤ 4.0 (very large) and *d* > 4.0 (extremely large). Moreover, upper and lower 95% confidence intervals of the difference (95% CId) were calculated. The statistical significance level was set at *p* < 0.05.

## 3. Results

The HRmax value recorded during the multistage 20 m multistage shuttle run test was 194 ± 12 beats·min^−1^. Table 1 presents the physiological responses and perceived exertion during different combat sessions. Regarding HR before combats, as expected, no significant main effects of training type (*F*_2,33_ = 2.07, *p* = 0.142) and area (*F*_2,66_ = 0.12, *p* = 0.88) were found, and there was no significant interaction between training type and area (*F*_4,66_ = 0.80, *p* = 0.531). For HRmean, there was no significant main effects of training type (*F*_2,33_ = 3.01, *p* = 0.063) or area size (*F*_2,66_ = 1.18, *p* = 0.314) and no interaction between training type and area was found (*F*_4,66_ = 2.34, *p* = 0.064). Similarly, for %HRmax, there was no main effects of training type (*F*_2,33_ = 0.619, *p* = 0.540) or area size (*F*_2,66_ = 0.287, *p* = 0.751), and no interaction between training type and area was found (*F*_4,66_ = 0.398, *p* = 0.310). However, there was a main effect of training type for HRpeak (*F*_2,33_ = 6.63, *p* = 0.004). Specifically, the Bonferroni post hoc test showed lower values in ne-waza compared to free randori (95% CId = −18; −2, *d* = −0.63 (moderate), *p* = 0.022) and tachi-waza (95% CId = −21; −5, *d* = −0.78 (moderate), *p* = 0.005). There was also a main effect of area size (*F*_2,66_ = 3.41, *p* = 0.039), with Bonferroni post hoc test showing lower values in 8 × 8 m compared to 4 × 4 m area (95% CId = −18; −1, *d* = −0.54 (small), *p* = 0.035). However, no interaction between training type and area size was found (*F*_4,66_ = 2.24, *p* = 0.074).

Regarding blood lactate concentration values after sessions, a main effect of training type was found (*F*_2,33_ = 8.67, *p* = 0.009), with Bonferroni post hoc test showing higher values during free randori compared to tachi-waza (95% CId = 1.5; 4.4, *d* = 0.97 (moderate), *p* = 0.008) and ne-waza (95% CId = 2.04; 4.9, *d* = 1.15 (moderate), *p* = 0.002). Likewise, a main effect of area was also observed (*F*_2,66_ = 13.50, *p* < 0.001), with Bonferroni post hoc test showing higher values for 4 × 4 m compared to 6 × 6 (95% CId = −0.01; 2.85, *d* = 0.47 (small), *p* = 0.047) and 8 × 8 m (95% CId = 1.4; 4.6, *d* = 0.88 (moderate), *p* < 0.001) areas. Nevertheless, no interaction was found between training type and area size (*F*_4,66_ = 1.57, *p* = 0.193). Regarding RPE, there were no main effects of training type (*F*_2,33_ = 1.53, *p* = 0.231) or area size (*F*_2,66_ = 3.12, *p* = 0.051), and no interaction between training type and area (*F*_4,66_ = 1.09, *p* = 0.368) was detected.

## 4. Discussion

The present study investigated the potential combined effect of area size (i.e., 4 × 4, 6 × 6, and 8 × 8 m) and training type (tachi-waza, ne-waza, and free randori) on the physiological responses and perceived exertion in female judo athletes. The main finding was that [La] was higher after free randori compared to tachi-waza and ne-waza and higher in 4 × 4 m compared to the other areas. This supports our first hypothesis, at least partially (because HR responses were not fully consistent with [La]). Rating of perceived exertion scores remained unchanged, confirming our second hypothesis.

Our results showed that [La] increased significantly after all combat sessions with values ranging from 7.85 ± 2.85 to 14.3 ± 2.2 mmol·L^−1^. In this consideration, Sbricolli et al. [17] showed that [La] values recorded after female judo combats were 9.2 ± 2.0 mmol·L^−1^, slightly lower than those recorded in almost all combat sessions of the present study. Similarly, with a limited number of participants (i.e., only two judo female athletes), Lehman [18] reported that [La] values were 8.47 ± 2.8 and 10.29 ± 3.72 mmol·L^−1^ during national and international judo championships, respectively. Such values are comparable to those reported in adult judo athletes executing different types of randori [6]. Generally, these results indicate that the combinations between varied area size and training modes resulted in physiological stress similar to [19] or higher than those observed in previous simulated competitive environments [17,20,21]. As stated above, [La] post combat was higher in free randori than tachi-waza and ne-waza. Indeed, this would reflect that the glycolytic system resulted in more stress during free randori compared to the other training modalities. It was reported that [La] increase was positively associated with the number of attacks [22] and that the type of throwing technique could affect the total energy expenditure, with higher energy cost for arm technique compared to leg technique [21]. Although not assessed, it is possible that our participants executed a higher number of attacks during the free randori, inducing higher metabolic solicitation compared to tachi-waza and ne-waza [23]. However, it is worth noticing that the higher [La] after free randori compared to the other two combat sessions is not in agreement with the results reported by Sikorski [8], who found higher [La] after standing position (11.3 ± 3.6 mmol·L^−1^) compared to groundwork randori (7.7 ± 2.2 mmol·L^−1^). To shed light on this, future studies should determine the time motion during these training types to allow better understanding of the aspects generating specific physiological responses. Regarding the effect of varied area size, [La] post combat was greater in 4 × 4 m compared to other areas. Similar results were reported by Ouergui et al. [12], i.e., 4 × 4 m area size caused greater [La] compared to 6 × 6 and 8 × 8 m area in female judo matches. This result can be explained by the fact that a reduced area size induces more grip disputes and shorter time spent in displacement without contact [13], consequently inducing higher [La] [2].

On the subject of cardiovascular responses, the current research indicated that HRmean values in different match sessions spanned from 154 ± 5 to 161 ± 7 beats·min^−1^. When intensities were expressed as %HRmax recorded during different experimental conditions, they varied from 79% to 83% of HRmax, which is lower than previously recorded value during simulated female judo combats, with intensity reaching 93% of HRmax [17]. Despite similarity regarding the total combat duration, this discrepancy may be attributed to differences in the activity profile (i.e., duration of efforts and pauses) between these combat sessions and those of the previous study [17]. Additionally, HRmean and %HRmax values did not change over different area sizes and training types. These results agree with those of a previous study in taekwondo [11], which showed that area size was not effective in altering cardiovascular responses during simulated tasks. However, as judo match is not performed at maximum intensity throughout its duration [2], only the peak HR was affected by the training type and area size, with values ranging from 162 ± 33 to 188 ± 15 beats·min^−1^. It has been reported that HRpeak during judo contests ranged from 190 to 200 beats·min^−1^ in male and female judokas [24]. We found that lower values of HRpeak were recorded in the ne-waza condition compared to the free randori and tachi-waza conditions and in 8 × 8 m compared to 4 × 4 m area. It has been suggested that displacement over extended trajectories can alter cardiovascular responses [11]. Such differences in HRpeak between training type can also be attributed to the time structure through these conditions as a lower rest interval results in higher HR values [25]. Likewise, we found lower HRpeak values in ne-waza compared to tachi-waza. This can be explained by the fact that, in groundwork combat, one of the athletes remains in a prone position held by the opponent, which probably limits the HR from reaching peak values. Thus, it can be suggested that combat-based exercises, mainly free randori and tachi-waza, are the most suitable training modalities to elicit cardiovascular responses closer to those recorded during judo combats and can be more appropriate for specific judo conditioning. Indeed, Olympic-level judo athletes have reported practicing more of the tachi-waza version followed by complete version and groundwork version responses [12]. Moreover, Olympic-level judo athletes reported spending 63 ± 37 min in the tachi-waza version but similar durations in the groundwork (36 ± 20 min) and complete versions (35 ± 31 min) in a typical judo session [6]. However, a study investigating the average HR response during only standing or groundwork bouts (four 4 min bouts interspersed with 1 min intervals) in young judo athletes reported no significant differences between these conditions [9]. It is worth assuming that HR values would have been different if male athletes were being evaluated as they are known to show a different cardiovascular response to exercise [26].

Another issue to discuss is the perceived exertion. Although the RPE scores seem to have a strong relationship with [La] [27] and HR [20] during judo matches, RPE scores in the present study were not different between training types, with scores ranging from 4 ± 1 to 6 ± 1. Referring to male judo combats, it was reported that the intensity of combats based on RPE scores was close to “very hard” after simulated [20] and official judo combats [28]. Similarly, Ouergui et al. [10] reported that altering area size was not effective in modifying RPE scores in kickboxing sparring sessions. Likewise, Ouergui et al. [11] showed that RPE responses in adolescent taekwondo athletes did not vary when area size was manipulated. The absence of difference in RPE scores may reflect the ability of judo athletes to regulate their actions regardless of their fatigue status [13]. Contrarily, these findings are not in agreement with those indicated by Ouergui et al. [12] in female judo athletes, where RPE scores were greater in 4 × 4 m compared to 6 × 6 and 8 × 8 m areas. This difference could be related to the variation in the effort-to-pause ratio across combats as this variable has the most important effect on combat intensity [7].

Some limitations in the present study should be acknowledged. First, the relationship between technical actions and physiological responses was not investigated. Such relationship could help coaches better understand how judo-specific actions are related to fatigue. Moreover, time-motion analysis could give information about the temporal structure of each training type and could give explanations regarding the results found. Therefore, physiological responses may be coupled with an accurate time-motion analysis aimed at finding whether physiological responses are related to technical–tactical skills in judo. Another limitation of the present study was the lack of reliability assessment of the measures (due to limited availability of the athletes). Finally, further cardiac stress outcomes (e.g., troponin T (TnT)) could be investigated within the present study’s context given that it is known TnT release is related to exercise stress intensity in athletes [29].

## 5. Conclusions

In conclusion, the present study showed that HRpeak was higher in tachi-waza and free randori compared to ne-waza and lower in 8 × 8 m compared to 4 × 4 m area. Moreover, [La] post combat was greater after free randori compared to tachi-waza and ne-waza and in 4 × 4 m compared to 6 × 6 and 8 × 8 m areas. These results show that all the experimental conditions investigated here can be manipulated to develop cardiovascular and anaerobic fitness. In fact, anaerobic conditioning can be developed by reducing the area size and using free randori as [La] was higher in these conditions. Moreover, these results indicate that specific training programs should be designed to include exercises stimulating anaerobic and aerobic metabolism simultaneously. Coaches aiming to prepare athletes always try to handle the demands of competition. To do this, coaches can administer free randori training in a small combat area to increase training intensity. Increasing training intensity is aimed at making the demands of subsequent competition more tolerable and less burdening, i.e., allowing athletes to handle their matches more easily.

## Figures and Tables

**Table 1 ijerph-19-03457-t001:** Lactate, heart rate, and rating of perceived exertion for different judo female combat conditions (values shown as mean ± standard deviation; *n* = 12).

	[La] Pre (mmol/L)	[La] Post (mmol/L)	HRpre (Beats/min)	HRmean (Beats/min)	HRpeak (Beats/min)	HRmean (%HRmax)	RPE
			Free randori				
4 m × 4 m	2.9 ± 1.1	14.2 ± 2.2 ^a,b,c,d^	98 ± 15	161 ± 7	184 ± 10	83 ± 6	7 ± 1
6 m × 6 m	2.9 ± 1.0	12.8 ± 2.2 ^a,b^	101 ± 26	161 ± 6	183 ± 6	83 ± 4	6 ± 1
8 m × 8 m	2.9 ± 0.8	12.9 ± 3.3 ^a,b^	99 ± 22	158 ± 8	181 ± 12 ^¶^	82 ± 6	6 ± 2
			Tachi-waza				
4 m × 4 m	2.8 ± 0.9	11.9 ± 3 ^c,d^	89 ± 16	157 ± 1	189 ± 15	81 ± 5	6 ± 1
6 m × 6 m	2.6 ± 1.1	10.9 ± 3.2	93 ± 20	154 ± 5	181 ± 8	80 ± 7	6 ± 2
8 m × 8 m	2.5 ± 0.7	8.2 ± 3.2	99 ± 18	160 ± 7	186 ± 3 ^¶^	83 ± 6	5 ± 1
			Ne-waza				
4 m × 4 m	2.3 ± 0.9	11.7 ± 3.1 ^c,d^	89 ± 14	160 ± 7	183 ± 9 *^,†^	83 ± 7	6 ± 2
6 m × 6 m	2.6 ± 0.9	9.9 ± 3.5	88 ± 22	157 ± 3	173 ± 12 *^,†^	81 ± 7	6 ± 2
8 m × 8 m	2.7 ± 0.9	7.9 ± 2.8	83 ± 24	157 ± 3	162 ± 34 *^,†,¶^	81 ± 5	5 ± 1

* Main effect of training type: HRpeak was lower in ne-waza compared to free randori (*p* = 0.002); ^†^ main effect of training type: HRpeak was lower in ne-waza compared to tachi-waza (*p* = 0.005); ^¶^ main effect of area size: HRpeak was lower in 8 × 8 m compared to 4 × 4 m area (*p* = 0.035); ^a^ main effect of training type: [La] post combat was greater in free randori in comparison to tachi-waza (*p* = 0.008); ^b^ main effect of training type: [La] post combat was greater in free randori in comparison to ne-waza (*p* = 0.002); ^c^ main effect of area size: [La] post combat was greater in 4 × 4 m compared to 6 × 6 m area (*p* = 0.047); ^d^ main effect of area size: [La] post combat was greater in 4 × 4 m compared to 8 × 8 m area (*p* < 0.001). [La] = blood lactate concentration, HRmean = mean heart rate, HRpeak: peak heart rate, HRmax = maximum heart rate, RPE = rating of perceived exertion.

## Data Availability

The data presented in this study are available on request from the corresponding author.

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
