# Peer review of "The Role of Competition Area and Training Type on Physiological Responses and Perceived Exertion in Female Judo Athletes"

_ijerph, 2022, doi:10.3390/ijerph19063457_

Round 1
Reviewer 1 Report
The authors have addressed most of my comments satisfactorily. However, I consider salami-slicing a major flaw-- the new findings that the authors presented in this manuscript do not warrant publication in this journal, which has high impact.
Author Response
Does the introduction provide sufficient background and include all relevant references?
(x) Can be improved
Please read below comments to specific point.
Is the research design appropriate?
(x) Can be improved
Please read below comments to specific point.
Are the methods adequately described?
(x) Can be improved
Please read below comments to specific point.
Are the results clearly presented?
(x) Can be improved
Please read below comments to specific point.
Point 1: … However, I consider salami-slicing a major flaw -- the new findings that the authors presented in this manuscript do not warrant publication in this journal, which has high impact.
Response 1: We thank expert reviewer for her/his comment. In our previous response, we tried to address above issue. We do not have any further comment regarding this.
We hope that the manuscript has now reached the standard necessary for formal acceptance endorsement in International Journal of Environmental Research and Public Health.
We look forward to hearing from you.
Best regards
Reviewer 2 Report
Following the author's previous study about different area size and effort-to-rest ratio, in this manuscript, from another perspective, the authors investigated the combined effects of competition area size and judo-specific training type on physiological responses in female judo athletes.
Their results showed that HRpeak was higher in tachi-waza and free randori compared with ne-waza, and the HRpeak was smaller in the 8x8 m area compared to the 4x4 m area in ne-waza.
Blood lactate concentration (La) of post combat was greater in 4x4 m area compared to the 6x6 m and the 8x8 m areas, and after free randori was greater compared to tahi-waza and ne-waza
This study aimed to verify whether the area size and training type can affect the physiological responses and perceived exertion in female judo athelets.
The manuscript is easy to follow and the data convincing, the study will be useful to provide a scientific basis for the coaches to formulate the best training program.
Comments:
As the authors said in the discussion, this study was the lack of measures’ reliability assessment.
Author Response
… The manuscript is easy to follow and the data convincing, the study will be useful to provide a scientific basis for the coaches to formulate the best training program. …
Thank you very much for appreciating our work.
We hope that the manuscript has now reached the standard necessary for formal acceptance endorsement in International Journal of Environmental Research and Public Health.
We look forward to hearing from you.
Best regards
Reviewer 3 Report
Proposed paper is interesting and well written. I have only two minor point in order to implement the discussion:
- Do you think there could be different results evaluating males? In fact, a recent study found that women showed a lower response to cardiac rehabilitation when compared to men (i.e. High Blood Press Cardiovasc Prev. 2021 Nov;28(6):579-587.). Please comment in the discussion.
- Do you think that other measurements of cardiac stress (such as troponin T) could be used in this context and will give important results? in fact another study found TnT release to be related to intensity of the sport stress in cycling athletes (High Blood Pressure and Cardiovascular Prevention. 2018 Mar;25(1):89-96. Please comment in the discussion.
Author Response
Are the conclusions supported by the results?
(x) Must be improved
Please read below comments to specific points.
Point 1: Do you think there could be different results evaluating males? In fact, a recent study found that women showed a lower response to cardiac rehabilitation when compared to men (i.e., High Blood Press Cardiovasc Prev. 2021 Nov;28(6):579-587). Please comment in the discussion.
Response 1: We thank expert reviewer for her/his suggestion. Suggestion was operated as follows (and relative reference was added accordingly):
“It is worth supposing that HR values would have been different evaluating male ath-letes, known to show a different cardiovascular response to exercise [26].”
Point 2: Do you think that other measurements of cardiac stress (such as troponin T) could be used in this context and will give important results? In fact another study found TnT release to be related to intensity of the sport stress in cycling athletes (High Blood Pressure and Cardiovascular Prevention. 2018 Mar;25(1):89-96. Please comment in the discussion.
Response 2: Suggestion was operated as follows (and relative reference was added accordingly):
“Finally, further cardiac stress outcomes (e.g., troponin T, TnT) could be investigates within present study’s context given that it is known that TnT release is related to exercise stress intensity in athletes [29].”
We hope that the manuscript has now reached the standard necessary for formal acceptance endorsement in International Journal of Environmental Research and Public Health.
We look forward to hearing from you.
Best regards
Reviewer 4 Report
Dear authors,
the actual, corrected form of the planned publication "The Role Of Competition Area And Training Type On Physio-logical Responses And Perceived Exertion In Female Judo Athletes" is now on a good practical and scientific level. The arguments and analysis of intensifying specific training are clearly presented with appropriate statistical analysis.
Considering the context of improving female athletes, I am missing the value of athletic training esp. strength training in order to makes demands under competitive conditions more tolerable and less strenuous. This, of course is not in the focus of this publication but would definetely deliver a broader context to the competitive approach and could be easily integrated with few sentences.
Author Response
Point 1: Considering the context of improving female athletes, I am missing the value of athletic training esp. strength training in order to makes demands under competitive conditions more tolerable and less strenuous. This, of course is not in the focus of this publication but would definitely deliver a broader context to the competitive approach and could be easily integrated with few sentences.
Response 1: We thank expert reviewer for her/his suggestion. Suggestion was operated as follows:
“Increasing training intensity is aimed at making following competition’s demand more tolerable and less burdening, i.e., allowing the athletes to handle their matches more easily.”
We hope that the manuscript has now reached the standard necessary for formal acceptance endorsement in International Journal of Environmental Research and Public Health.
We look forward to hearing from you.
Best regards
Reviewer 5 Report
This study investigates the effect of training area and judo-specific training type on physiological responses and perceived exertion in female judoka.
Firstly, it is very unusual to receive a tracked copy for review. This manuscript should be presented clean.
Change "physio-logical" to "physiological" in the title.
Line 22: Change "Study investigated combined effects..." to "This study investigated the combined effects..."
LIne 22: Change "...8x8 m)/judo-specific..." to "...8x8 m) and judo specific..."
Lines 25-26: Do you mean sub-elite and elite athletes?
Line 27: The sentence beginning "Following measurement/time point..." is very confusing. Please rephrase it.
Line 29: Change "operated" to "used to compare between conditions"
Line 63: Change "coaches" to "coaching"
Line 72: Change "of" to "in"
Line 83: Change "compared to" to "compared with"
Line 93: delete "size"
Line 110: Change "resulted" to "was"
Line 125: Please provide the ethical approval number in brackets.
Line 127: Did the athletes only sign the form? or did their guardians and coaches also sign the form?
Line 134: Change "analysis" to "analyses"
Line 139: Add "were" before "familiarized"
Line 145 - 147: Did each athlete perform each session with the same training partner? Or did they have a different partner for each session?
Line 150: For readers unfamiliar with the judo scoring system, including a brief explanation could be included early in the Introduction section where you describe a typical judo match, around line 46.
The Results section is clearly presented.
Line 213: "confirms" is a little strong for this claim. I would change it to "This supports our first hypothesis, at least partially..."
Line 227: Change "environment" to "environments"
Line 299: Change "compared to" to "compared with". This difference in effort to pause ratio is an interesting variable.
The authors acknowledge that a time-analysis would add to the data. I agree and think that quantifying the number of movements, ie attempted throws in tachi-waza and pins in ne-waza, and both for randori would also be of interest to the reader and would strengthen the study. I am sure the authors will consider this for future studies.
Author Response
Point 1: Change "physio-logical" to "physiological" in the title.
Response 1: We thank expert reviewer for her/his suggestion. Yet, the title says “physio-logical” instead of “physiological” only because of the hyphenation rule featuring the journal’s Word template.
Point 2: Line 22: Change "Study investigated combined effects..." to "This study investigated the combined effects..."
Response 2: Sentence was changed following your suggestion.
Point 3: Line 22: Change "... 8x8 m)/judo-specific..." to "... 8x8 m) and judo specific..."
Response 3: Sentence was changed following your suggestion.
Point 4: Lines 25-26: Do you mean sub-elite and elite athletes?
Response 4: Yes. Sentence was clarified following your suggestion.
Point 5: Line 27: The sentence beginning "Following measurement/time point..." is very confusing. Please rephrase it.
Response 5: Sentence was clarified and now starts as follows:
“Following measurements at different time-points were chosen…”
Point 6: Line 29: Change "operated" to "used to compare between conditions".
Response 6: Sentence was changed following your suggestion.
Point 7: Line 63: Change "coaches" to "coaching".
Response 7: Sentence was changed following your suggestion.
Point 8: Line 72: Change "of" to "in".
Response 8: Sentence was changed following your suggestion.
Point 9: Line 83: Change "compared to" to "compared with".
Response 9: Sentence was changed following your suggestion.
Point 10: Line 93: delete "size".
Response 10: Sentence was changed following your suggestion.
Point 11: Line 110: Change "resulted" to "was".
Response 11: Sentence was changed following your suggestion.
Point 12: Line 125: Please provide the ethical approval number in brackets.
Response 12: Ethical approval number was provided in brackets.
Point 13: Line 127: Did the athletes only sign the form? or did their guardians and coaches also sign the form?
Response 13: All of them signed the form.
Point 14: Line 134: Change "analysis" to "analyses".
Response 14: Sentence was changed following your suggestion.
Point 15: Line 139: Add "were" before "familiarized".
Response 15: Sentence was changed following your suggestion.
Point 16: Line 145 - 147: Did each athlete perform each session with the same training partner? Or did they have a different partner for each session?
Response 16: Each athlete had a different partner for each condition. This information was added.
Point 17: Line 150: For readers unfamiliar with the judo scoring system, including a brief explanation could be included early in the Introduction section where you describe a typical judo match, around line 46.
Response 17: A judo scoring system brief explanation was added around original line 46. Ibrahim, do you agree?
Point 18: Line 213: "confirms" is a little strong for this claim. I would change it to "This supports our first hypothesis, at least partially..."
Response 18: Sentence was changed following your suggestion.
Point 19: Line 227: Change "environment" to "environments"
Response 19: Sentence was changed following your suggestion.
Point 20: Line 299: Change "compared to" to "compared with".
Response 20: We thank expert reviewer for his suggestion. Sentence was changed following your suggestion.
We hope that the manuscript has now reached the standard necessary for formal acceptance endorsement in International Journal of Environmental Research and Public Health.
We look forward to hearing from you.
Best regards
This manuscript is a resubmission of an earlier submission. The following is a list of the peer review reports and author responses from that submission.
Round 1
Reviewer 1 Report
The Authors decided to extend the subject of physiological response to physical activity in judo female athletes. The subject covered by the Authors is very interesting and could be useful for future training planning purposes, however I would like to express my concerns and doubts about the proposed manuscript.
The part of the results in the reviewed manuscript has been already published by the Authors in the other paper (impact of competition area - 4×4 m, 6×6 m, 8×8 m) and the only new parameters added to the analysis (impact of judo-specific training type - tachi-waza, ne-waza, free combat) were gathered at the same time during this study, and were already available at the time of the publication of the other paper. To conclude, all data gathered in the course of this study could have been published in one paper instead of dividing it into two with a smaller set of data. The Authors decided to describe the same study in two papers with the difference of only one parameter. This kind of practice is not acceptable in my opinion. Moreover, the only data that differs these two papers (impact of judo-specific training type) is not enough to be the basis of the paper published in a such high IF journal.
What is more, the Authors discuss their results mainly with the results from their own work with only few notes of papers prepared by other Authors. I suggest the Authors focusing more on the other Authors’ work and preparing longer list of literature to discuss with.
To sum up, for the purpose of the future publications I suggest that the Authors should publish all the related results from one study in one publication instead of dividing it into two or more publications with small differences between data.
Reviewer 2 Report
This paper examined the effect of competition area and training type on physiological responses and perceived exertion among female judo athletes. Some major issues raise significant concerns. In particular, the authors acknowledged that this study was part of a prior study (reference 11, Ouergui et al.) with the same sample of 12 participants. That study examined the effect of similar/ same independent variables on similar/ same dependent variables. It is unclear why current results were not reported in one paper. It would be important for the author to explain how this was not salami slicing, which is highly discouraged in scientific reporting.
Other major and minor comments:
- Title, abstract, and throughout the manuscript: Rather than using the term “perceptive responses” or “perceptual responses”, suggest to replace them with “subjective responses” or “perceived exertion” (as stated in the hypothesis). Perception may refer to visual or hearing perception, and therefore the term may cause confusion.
- METHODS: p.2, line 102 – 104: Can the authors clarify if sample size was estimated based on lactate or perceived exertion as the outcome measure? It could not have been both.
- METHODS: p. 3, line 125-6: Please provide more information about the rating of perceived exertion in this paper to make it easy for the readers. For instance, is the possible range 0 to 10? What does a rating of 0 refer to? What does a rating of 10 refer to?
- METHODS: p.3, lines 119-126, 135-136: The authors described some of the study procedures and referred readers to Ouergui et al. (11). They also stated that “All experimental sessions were randomly administered and ...” (lines 135-136). Please clarify what random administration means. Did each of the 12 participants undergo each and all of the 9 treatment conditions? If that was the case, what was done to address possible practice effect and carryover effect on the outcomes? Please clarify and elaborate.
- METHODS and RESULTS: How many 2-way ANOVA with repeated measures were conducted? It was unclear how many variables were used as the outcome measure in the analyses.
- RESULTS: Would it be possible for add a table of zero-order correlations among the outcome variables, including their pre and post values?
- RESULTS: Table 1” “a.u.” referred to arbitrary units of RPE according to the Table’s notes. What does that mean? The scale came from a published study and there were meanings attached to the ratings according to the study.
- DISCUSSION: P.6: Did the findings support, or refute, the 2 hypotheses? In the introduction, the authors hypothesized that smaller area and free randori would result in higher physiological strain, and that the rating of perceived exertion scores would remain unchanged. Please state which part was not supported right in the beginning of Discussion.
- Some grammatical and other language issues are highlighted below as examples. They make the manuscript difficult to read. Please check the whole manuscript and seek help from a professional editor as needed.
- p.2, lines 61-63: “In this regard, Sikorski [7], compared a 5-min judo match disputed in a standing position with a match of the same duration performed on the ground.”
- p.2, lines 70-71: “Authors showed increases of AST, CK and LDH concentration post both tachi-waza and ne-waza sessions but no difference between training methods suggesting similar muscle damage [8].”
- p.3, lines 139-140: “Before each randori, athletes performed a standardized warm-up session lasting 15 min made of jogging and dynamic stretching, with following 3 min of passive rest and it were performed baseline measures.”
-p.7, lines 287-288: “… to better prepare athletes to handle the competition demands, coaches can use here investigated judo-specific exercises that combine both technical-tactical and physical components,…”